# Health Outcomes of an Integrated Behaviour-Centred Water, Sanitation, Hygiene and Food Safety Intervention–A Randomised before and after Trial

**DOI:** 10.3390/ijerph17082648

**Published:** 2020-04-13

**Authors:** Tracy Morse, Elizabeth Tilley, Kondwani Chidziwisano, Rossanie Malolo, Janelisa Musaya

**Affiliations:** 1Centre for Water, Environment, Sustainability and Public Health, Department of Civil and Environmental Engineering, University of Strathclyde, Glasgow G1 1XJ, UK; kchidziwisano@poly.ac.mw; 2Centre for Water, Sanitation, Health and Appropriate Technology Development (WASHTED), University of Malawi (Polytechnic), Blantyre, Malawi; rossaniedaudi@yahoo.com; 3Department of Environmental Health, University of Malawi (Polytechnic), Blantyre, Malawi; Elizabeth.Tilley@eawag.ch; 4Swiss Federal Institute of Aquatic Science and Technology (Eawag), CH-8600 Duebendorf, Switzerland; 5Department of Biochemical Sciences, College of Medicine, University of Malawi, Blantyre, Malawi; jmusaya@medcol.mw

**Keywords:** food hygiene, behaviour change, water, sanitation and hygiene, low and middle income countries, diarrhoeal disease

## Abstract

Diarrhoeal disease in children under five in low income settings has been associated with multiple environmental exposure pathways, including complementary foods. Conducted from February to December 2018 in rural Malawi, this before and after trial with a control used diarrhoeal disease as a primary outcome, to measure the impact of a food hygiene intervention (food hygiene + handwashing) relative to a food hygiene and water, sanitation and hygiene (WASH) intervention (food hygiene + handwashing + faeces management + water management). The 31-week intervention was delivered by community-based coordinators through community events (*n* = 2), cluster group meetings (*n* = 17) and household visits (*n* = 14). Diarrhoeal disease was self-reported and measured through an end line survey, and daily diaries completed by caregivers. Difference-in-differences results show a 13-percentage point reduction in self-reported diarrhoea compared to the control group. There were also significant increases in the presence of proxy measures in each of the treatment groups (e.g., the presence of soap). We conclude that food hygiene interventions (including hand washing with soap) can significantly reduce diarrhoeal disease prevalence in children under five years in a low-income setting. Therefore, the promotion of food hygiene practices using a behaviour-centred approach should be embedded in nutrition and WASH policies and programming.

## 1. Introduction

Diarrhoeal disease continues to be a leading cause of mortality and morbidity in low and middle income countries, with 62.2% of diarrhoeal disease deaths in children under five attributed to poor water, sanitation and hygiene (WASH) and the associated consumption of contaminated foods in household settings [1,2,3]. A range of recent studies have examined our current understanding and the efficacy of control measures taken to reduce exposure to enteric pathogens, including attempts to introduce multiple barriers to exposure pathways: enclosed sanitation, household water treatment, hand washing with soap, clean play spaces, domestic animal control and hygiene of complementary foods [4,5,6,7,8,9,10,11,12,13,14,15,16,17,18]. 

These multiple environmental pathways of transmission in low income countries make it difficult to attribute disease to a specific intervention, leading to calls for interventions that target the reduction of faecal contamination across the domestic environment, and develop a more appropriate package of effective WASH interventions [8,19,20,21,22]. Despite the clear need for reducing faecal-oral contact, less than 5% of the population of Sub-Saharan Africa have access to improved water, sanitation and hygiene as described by the Sustainable Development Goal indicators [23]. These complex and detrimental problems need a more integrated approach taking into consideration both widespread exposure to faecal contamination [21,24,25], and the social, technological and economic influences that promote participation in and the uptake of sustained behavioural changes related to WASH at the household level [8,26,27,28,29,30]. The use of over-simplified interventions that have historically ignored these complexities is one potential reason why progress has been slow.

The Hygienic Family (*Banja la Ukhondo*) intervention was based on formative research that found that children under the age of two were being exposed to multiple sources of faecal contamination within their domestic environments, particularly through complementary feeding at shared family meals, household drinking water, and exposure to environmental faeces [31,32,33]. However, as previous studies addressing food hygiene in low income settings measured success based on behavioural change, structural changes or the microbiological quality of food and water [5,9,10], our study was designed to measure the relative impact of the hygiene of complementary foods on diarrhoeal disease as a primary outcome, and the impact of a food hygiene intervention (food hygiene + handwashing) relative to a food hygiene and WASH intervention (food hygiene + handwashing + faeces management + water management) [31]. 

## 2. Materials and Methods

### 2.1. Background and Study Design

As described previously [31], the study was a randomised before and after trial, with a control and two treatment arms, assessing the relative impact of the hygiene of complementary foods on the prevalence of diarrhoeal disease in children under five years old. For the programme, implemented from February–November 2018 in the rural District of Chikwawa in Southern Malawi, we recruited 1000 households in total. The two treatment arms were designed based on formative research: (1) a food hygiene intervention only (*n* = 400), and (2) an integrated WASH and food hygiene intervention (*n* = 400) [31], with a control group in which no intervention took place (*n* = 200). Although diarrhoeal disease was the primary outcome, respiratory infections were also monitored as indicators of improved WASH and food hygiene practices. End line data were collected in December 2018.

### 2.2. Study Area and Participant Selection 

Malawi is divided into 28 Districts, which are subdivided into Traditional Authorities (TAs). Each TA contains villages, which are administered by chiefs and/or village heads. There are 12 Traditional Authorities (TAs) within Chikwawa district. This study was based in four different TAs: one in Treatment 1, two in Treatment 2, and one in the control. Treatment 2 included two TAs as there were insufficient eligible participants in the planned TA. Baseline data reported elsewhere showed homogeneity across all study groups in terms of household demographics, socio-economic status and hygiene related proxies [31]. All study sites were selected in collaboration with the District Coordinating Team (government intersectoral team) and had all been declared open defaecation free by the Government of Malawi. 

All recruited households had a functioning latrine and resided within 500 m of a functioning borehole, to ensure that there were no significant variations in access to water or sanitation infrastructure. Eligible households had a child aged between 4 and 90 weeks at enrolment (March–April 2017) to ensure that children were not neonates and that all children were under 60 months old at the end of intervention period (November 2018). The age of children was verified through birth and/or immunisation records supplied by the caregiver. All children in the target age range from eligible households were recruited at baseline; however, only one child per household was recruited. Physical recruitment was conducted by trained research assistants with the approval and support of community health workers (Health Surveillance Assistants (HSAs), traditional leaders (village chiefs) and community coordinators (female community members employed by the study). Morse, et al. reported the sample size calculations for and population demographics of the study sample [31]. 

Households were allocated to clusters (treatment areas *n* = 20; control area *n* = 10) of between 15 to 25 caregivers for the purpose of the intervention delivery. Due to the close proximity of these clusters to one another, and therefore the risk of spillover effects, the study was randomized by TA. Each TA was provided with a dedicated treatment arm coordinator to oversee the intervention delivery and data collection. 

Participants were advised that they were taking part in a study focused on child caregiving practices during consent, to minimise the reflexivity and reporting bias of health outcomes. 

### 2.3. Intervention

Based on the findings of the formative and baseline data, and using a behaviour-centred approach, the 31-week intervention included four critical areas as outlined in Figure 1 and described elsewhere [31]. 

The intervention was designed in modules (*n* = 4) to be delivered through community coordinators with the support of government HSAs, and oversight from study coordinators, to determine if the content and approach were appropriate for scaling up within existing structures. The content was conveyed through community-level open days at the beginning and end of the trial (*n* = 2). Additional content was delivered through cluster meetings (up to 25 caregivers per group) (*n* = 17) and household visits (one-to-one meetings) (*n* = 14), which took place on alternating weeks. Cluster meetings included community coordinator-led discussions, practical exercises, demonstrations, games and celebrations of success to promote behavioural change and build supportive networks and social capital within the group [31]. Household visits allowed community coordinators to assess progress, helped to put lessons into action, and provided support for any challenges that the households might have been facing. The number of cluster meetings and household visits varied depending on the material included in each topic. 

Both community coordinators and health workers were supported and supervised by treatment arm coordinators to ensure the integrity and fidelity of the content delivered. Each module of the intervention was preceded by a one-week course of training for the community coordinators and HSAs. Completion of the module was also followed up with a review exercise to evaluate the successes and challenges encountered, and outline proposed changes in the content or delivery mechanisms. 

### 2.4. Data Collection

Two health outcomes were measured in target children:Primary health outcome—diarrhoeal disease, defined by WHO as the passage of three or more loose or liquid stools per day [34].Secondary health outcome—acute respiratory infections, defined as children having a fever plus either a cough or nasal congestion, or a fever plus breathing difficulty [35].

All health outcomes were measured in two ways:Episodes of diarrhoea and acute respiratory infections were self-reported daily using household diaries. Diaries were checked for completeness during household visits (fortnightly) and collected monthly from February to December 2018 from caregivers by community coordinators. Diaries were collected from the field and recorded in Microsoft Excel for analysis.Episodes of diarrhoea and acute respiratory infections were self-reported for the two-week period preceding baseline (March 2017) and end line (December 2018) surveys. These surveys were conducted by trained research assistants, and the responses were recorded on tablets using the Kobo Collect software [36], cleaned and prepared for analysis.

### 2.5. Data Analysis 

The use of two treatment arms allowed us to measure:The impact of each intervention on the incidence of diarrhoeal disease. The impact of each intervention relative to the control group.The impact of the food hygiene intervention relative to the WASH + food hygiene intervention.

The primary variable of interest was the occurrence of diarrhoea within the past 2 weeks as reported by the primary caregiver, measured as a binary outcome. The impact of the interventions was analysed using a difference-in-differences approach, i.e., the prevalence of diarrhoea in the treatment areas was measured between baseline and the follow-up surveys and compared to that at the same time points in the control areas (*n* = 1000). Attendance at cluster meetings (individual level), cluster attendance score (cluster level), socio-economic characteristics (individual level), and household hygiene improvements (e.g., installation of a dish rack) (individual level), were included as covariates in the model. The incidence of secondary outcomes of acute respiratory infection was likewise analysed. 

Daily diarrhoea and respiratory infection values (continuous data) were collected as a way of monitoring temporal changes that may have resulted due to other interventions, environmental changes, seasonality, or other unknown factors. 

In the treatment areas, the attendance of caregivers at cluster meetings was measured using a register maintained by the community coordinators and checked by the treatment arm coordinators. Household visits were measured through the submission of completed household checklists completed by the community coordinators at each household visit and checked by the treatment arm coordinators. All attendance and visit records were consolidated by the treatment arm coordinators and submitted as weekly reports. 

Data were cleaned and analysed using Stata version 13.1 (College Station, TX, USA).

### 2.6. Ethics

The study protocol was approved by the University of Malawi’s College of Medicine Research Ethics Committee. Permission was obtained from the local authorities, those are, the Chikwawa District Council, the Chikwawa District Health Office, and the traditional chiefs. The participants were advised that they had the freedom to refuse participation or withdraw from the study at any time. Participants’ written informed consent was obtained before inclusion in the study. Participants were provided with a unique identifying number, and data were anonymized during data analysis. Data were accessed only by the authors. The study was registered with the Pan African Clinical Trials Registry (PACTR201703002084166). 

## 3. Results

### 3.1. Descriptive Statistics

The characteristics of participants for the study have been published elsewhere [31]. A total of 1000 children from separate households were recruited (Treatment 1 = 400, Treatment 2 = 400, Control = 200) as outlined in Figure 2. By end line, 18.7% of households had been lost to attrition, primarily due to relocation from the study area (Treatment 1 = 19.25%, Treatment 2 = 23.5%, Control = 8%) (Figure 2). At baseline, child level covariates such as age, gender, vaccinations, and the prevalence of diarrhoea and respiratory infections were similar at baseline between the treatment and control groups. Indicators for breastfeeding, the consumption of solid foods and drinking water, and the hygiene proxies for sanitation and hygiene in the home were also consistent across the control and treatment areas, with all households having a latrine [31].

Throughout the study, the proportions of children within specific age categories (months) were consistent across the two treatment and control areas (Figure 3). As shown in Figure 3, the entire cohort of children that were in the <6, 6–12, and 12–23 month age categories graduated into the 24–35 and 36–47 month categories except for one child who remained in the 12–23 month age category; this child was from Treatment 1.

### 3.2. Hygiene Proxies

Chi-square analysis was used to test whether there were significant differences between the differences in the intervention and control groups at baseline and follow-up: [follow-up−baseline] _intervention_-[follow-up−baseline] _control_. When sanitation and hygiene proxy measures (i.e., the presence of soap in the household, the presence of a handwashing facility, the presence of a dish rack, the presence of soap and water at a handwashing facility and at a utensil washing location) were compared between baseline and follow-up surveys across all of the three study areas, there were significant differences (*p* < 0.0001) both between the control and Treatment 1 (Table 1) and between the control and Treatment 2 (Table 2). However, when baseline and follow up data were compared between Treatment 1 and Treatment 2 (Table 3), the differences were not significant. Nevertheless, there were slightly more hygiene proxy measures measured in Treatment 1. For instance, 84% of the handwashing facilities in Treatment 1 had soap and water compared to 70% measured in Treatment 2.

### 3.3. Intervention Fidelity

As outlined in Table 4, the average attendance values at cluster meetings give an indication of the general participation rates. On average, these were at least 50% of the target number, though never close to 100%, indicating that there were consistent absences throughout the module delivery. Almost consistently, the average attendance was higher in Treatment 2 than in Treatment 1. The only systematic difference between these two groups was the treatment arm coordinator that delivered the intervention training and conducted supervisory visits to community coordinators during cluster meetings and household visits. We can therefore only assume that this unobserved factor played a role in the attendance of the caregivers. The percentage of the participants that attended no meetings (% none attended) or had no household visits indicate a monitoring gap on the part of the project implementors, and again, these values are consistently higher in Treatment 1. Similarly, the percentage of caregivers that attended all meetings or received all of the target household visits was higher in Treatment 2, reaching a maximum of 89% for the faeces management cluster meeting, and except for the visits pertaining to food safety and hygiene, the percentage that attended all household visits was consistently over 80%.

### 3.4. Health Outcomes

To determine the impact of the intervention, we collected self-reported illness (diarrhoea and respiratory infection) data two weeks prior to baseline or end line. As outlined in Figure 4, the percentages of children with diarrhoea were generally consistent, with no statistically significant differences between the groups at baseline, and they were in line with national averages. Throughout the period of the study, children moved into higher age categories, and the control group’s reduction in illness reflects a natural decline in line with national averages; however, illness in the intervention groups was seen to decrease further.

The results were further analysed using a difference-in-differences analysis and are presented in Table 5. At baseline (before the intervention), there were no significant differences between the Treatment and Control values for any of the outcome variables (diarrhoea and respiratory infection). There were significant differences between each Treatment area and the Control area at end line (after the intervention) for diarrhoea, but for respiratory infections, only between Treatment 2 and Control. The final difference in differences results were strong and significant for diarrhoea in both areas, and for respiratory infections in Treatment 2.

The impact on diarrhoea from the full set of interventions (four modules) in Treatment 2 was slightly higher (0.5%) than that from the two modules delivered in Treatment 1, though not sufficiently so that the additional 9 weeks of implementation could be considered efficient. However, the additional module delivery (faeces management and household water management) significantly reduced the incidence of respiratory infections (by almost 13 percentage points). Individuals responsible for delivery and monitoring appear to have had a non-measurable effect on the participants’ learning and retention. 

In order to better understand the other factors responsible for the measured outcome results, we constructed linear probability models for both of the binary outcome variables (diarrhoea and respiratory infection) at end line, including relevant explanatory variables. The results are shown in Table 6. Logistic regression estimations, shown as relative risk values, are also available in Appendix B. 

For diarrhoea in both treatment arms, none of the explanatory variables were significant in explaining the reported end line values; in other words, only the intervention, irrespective of any other household conditions or socio-demographic factors, was responsible for the prevalence observed (columns 1 and 3); even the numbers of visits or meetings that the caregivers engaged in were not responsible. On the other hand, the actual intervention (columns 2 and 4) was highly significant in predicting the reduction of diarrhoea. This paradox points to other factors that affected the behaviour observed, i.e., being in the treatment arm had an impact, but that impact was not dependent on the level of participation. However, the model fit was poor, indicating a large amount of unobserved variation. Due to the multifaceted nature of the intervention, it could be argued that the reduction was therefore associated with the impact of a high number of marginal effects from multiple sources rather that one overriding effect from a single change in behaviour. It is also likely that the qualities and personal characteristics of the individuals implementing the interventions may have played an important role in explaining the results observed. 

The results for respiratory infections are similar in that both interventions had significant impacts (columns 6 and 8), though unlike the results for diarrhoea, the reduction observed was significantly affected by both the baseline values for respiratory illness (columns 5–8) and the number of visits and meetings attended (columns 5 and 7).

While the results shown in Table 5 convey the net changes in the outcome indicators between the baseline and end line surveys, the results presented in Figure 5 illustrate the monthly changes that were recorded continuously between those points. These continuous data were self-reported by the caregivers on a daily basis, which were checked by the community coordinators who visited the household on a fortnightly basis and subsequently collected at the end of each month. Each household diary was specific to the target child in the study and recorded the number of episodes the child experienced for the outcome of interest. The results from the household daily diaries correlated with those of the self-reported incidences at baseline and end line. 

Data are plotted for a child having at least one incident or infection during the month (Figure 5a,b). All health outcome graphs reflect a challenge with data from the control group occurring in the months of April and July (2018), reflective of missing diaries misplaced by the group coordinator.

The results in Figure 5a highlight that although the number of children who had diarrhoea within a month were not significantly different between the control and treatment areas, the number of cases per month was higher in the control area and remained high over the period of measurement, while the values in the treatment areas were effectively null by the end of the period. 

Compared to diarrhoea, respiratory infection (Figure 5b) was much more common and affected nearly one quarter of the Treatment 1 population in March, which may be attributed to the hot and rainy season, which sees a proliferation of febrile illness and additional time spent in enclosed spaces due to wet weather. A later peak in respiratory illnesses would be associated with winter months in May to August. National figures show an average of 5.4% of children under five having acute respiratory illness (ARI), peaking at over 6% between 6 and 35 months and then declining to around 4.5%. Therefore, as with diarrhoeal disease, the decline seen across groups could be attributed to the increasing ages of the participants. However, as with diarrhoea, there was also a higher reduction in the number of reported cases of ARI in the treatment areas within a given month compared to in the control, inferring that improved hygiene practices may have led to an increased reduction in target households. 

## 4. Discussion

This before and after trial with a control in rural Malawi measured the relative impact of a multi-faceted hygiene intervention on the reduction of diarrhoeal disease in an under five population. Self-reported results indicated a significant reduction in diarrhoeal disease associated with improved hand washing with soap at critical times and food hygiene practices. Those households that also improved faeces and water management additionally showed a significant reduction in acute respiratory infections. We did not find any significant explanatory variables for the reduction in illness in the treatment arms, indicating that the gains were likely to be from marginal effects across multiple sources rather than from one over-riding single behaviour. This finding is in line with the outcomes in the recent body of evidence from a range of WASH trials, which have highlighted the need to tackle multiple routes of pathogen exposure and take a new approach to how these are addressed [8,19]. Although there may be concerns related to self-reporting illness, our study relied on both daily diaries and two week recall at baseline and end line. The data sources show similar reductions in disease prevalence across the study period, thereby supporting our reported findings. 

Minimal relative reductions in diarrhoeal disease between Treatments 1 and 2 infer a significant role played by food hygiene practices and associated hand washing with soap at critical times. The role of food in diarrhoeal disease transmission has been long recognised [37,38,39,40,41], and a recent WHO report attributed 70% of the burden of foodborne disease to low and middle income countries, with 40% of this burden affecting children under the age of five [2]. Our findings support the growing body of evidence that improving food hygiene practices can play a significant role in reducing diarrhoeal disease in under five populations. We also recognise the role that effective hand washing with soap may be playing in this reported reduction in diarrhoea, and acknowledge that even increased hand washing with water alone may be contributing to these improved health outcomes [18,42,43,44]. 

Similar reductions in diarrhoeal disease between Treatments 1 and 2 may also have been affected by the content of the WASH based modules: our study did not promote or subsidise point of use water treatment, which may have limited the impact of the water management improvements due to the continued post collection contamination of drinking water with *E. coli*. In addition, it may be indicative of the wide level of faecal contamination already present in the environment [13,14], which existed before the intervention supported the management of animal and child faeces from the household yard. 

In terms of reductions in respiratory infections, results infer the need for a wider approach to reducing environmental exposure to faecal matter. Our observed reduction associated with improved WASH practices is in agreement with previous reports of improved hygiene having a secondary impact on acute respiratory infections [35,45].

Our behaviour-centred intervention used the Risks, Attitudes, Norms, Abilities and Self Regulation (RANAS) approach, [46] and the embedded recommended behavioural change techniques [31], within existing community structures to provide both familiarity and innovation in how messages were delivered. Although the contact time was intensive, the mode of delivery used reflected that of existing community-based women’s and caregiver groups [47]. Our formative research outlined the challenges this approach may have regarding caregiver availability, given that the majority of households were subsistence farmers and, therefore, self-employed [31]. As such, the intervention content was designed in such a way as to provide repetition and ensure that missed cluster meetings or household visits would not impede long-term progress. The achievement of health outcome improvements in this study appears to show that this intensive and repetitive approach warrants further use, as attendance at neither cluster meetings nor household visits ever reached 100%. However, despite this, hygiene proxies and behaviours [48] were seen to significantly improve in the treatment versus control arms. Any intervention must cognisant of the context in which it is set and the other household commitments of caregivers and targeted populations, which may impact their ability to participate in community or household meetings. Therefore, by using group and one-to-one household meetings to provide the same message through a range of behavioural change techniques and concurrently build trust and social capital [49], such an approach could have effective implications for wider community hygiene improvements. This finding agrees with recent reviews which reported the benefits of a high degree of contact time to achieve the anticipated behavioural change, particularly at the household level [8,30]. 

The use of community-based coordinators and community health workers (HSAs) was tested in this intervention to determine if the low education level of coordinators could be overcome with effective training and supervision. Although we do acknowledge a degree of heterogeneity in the delivery and participation with the intervention content, the successful health outcomes of the intervention infer that the use of community-based coordinators, similar to that of care group coordinators (Scaling Up Nutrition) or maternal health women’s group coordinators [47] could be an effective route for implementation. The use of local and familiar facilitators can ensure an ongoing empathy with caregivers, as they have a working knowledge and personal experience of caregiver capabilities, pressures and stressors, all of which can affect a caregiver’s ability to make sustained structural and behavioural changes [50]. This is particularly important when considering the poorest populations and lowest levels of caregiver education in high risk populations in low income settings, where access to health care providers is limited, and exposure to faecal pathogens is high [1,51,52]. 

The modules developed through the Hygienic Family intervention have been designed in such a way as to support the integration of the content with other programmes, and to take into consideration the limitations of facilitators and irregular attendance from caregivers. There may be concerns that the integration of too many health and hygiene messages within specific caregiver groups could be overwhelming for both facilitators and participants. However, the call for an improved integration of services, as well as evidence from recent studies [53], demonstrates that such integration is an effective route for improving hygiene practices in the lowest quintile of caregivers by education.

The study limitations must be taken into consideration in the interpretation of these results. The intervention was limited to a nine-month period but likely would have been more effective over a full twelve months. Despite this, the results reflect a range of seasons within Malawi, including the initiation of the rainy season (November 2018), which sees the peak prevalence of diarrhoeal disease in the under-five population. End line data were also collected soon after (December 2018) the completion of the intervention (November 2018), which may have led to reflexivity in the responses. As outlined above, we believe that we have compensated for this with the use of two methods for reporting illness. We also acknowledge that our study did not measure all variables that may affect the prevalence of diarrhoea or respiratory infections, such as low birth weight status, malnutrition, HIV status, or the presence of other diseases such as measles, malaria, etc.

## 5. Conclusions

We conclude that food safety and hygiene interventions with embedded hand washing with soap at critical times can have a significant effect on the reduction of diarrhoeal disease in children under five years in low income settings. As such, the promotion of food hygiene practices using a behaviour-centred approach should be integrated into nutrition (e.g., Scaling Up Nutrition) and WASH (e.g., Community Led Total Sanitation) intervention policies and programming. Although limited benefits were seen from the addition of faeces and water management interventions, it must be considered that these may have been impacted by the existing environmental contamination in the household yard. Integration can be achieved through existing structures using locally available expertise with appropriate support and supervision.

## Figures and Tables

**Figure 1 ijerph-17-02648-f001:**
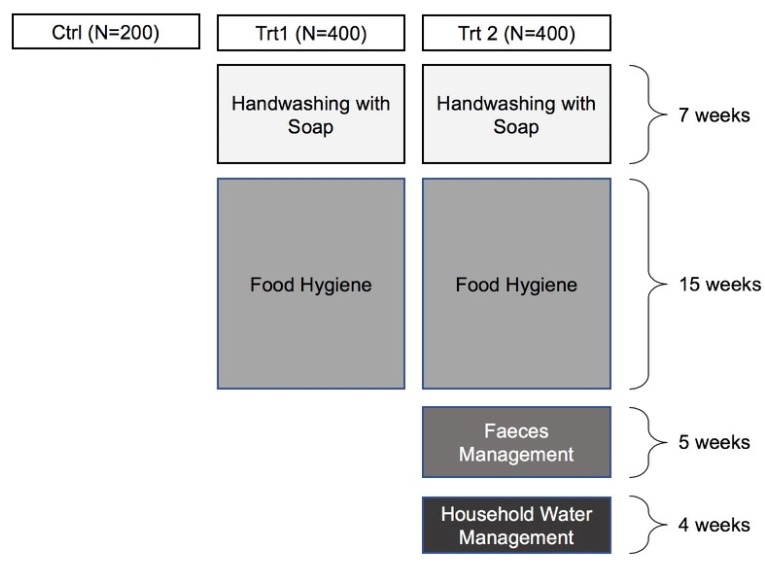
An outline of the intervention modules delivered within two treatment arms over the 31-week programme.

**Figure 2 ijerph-17-02648-f002:**
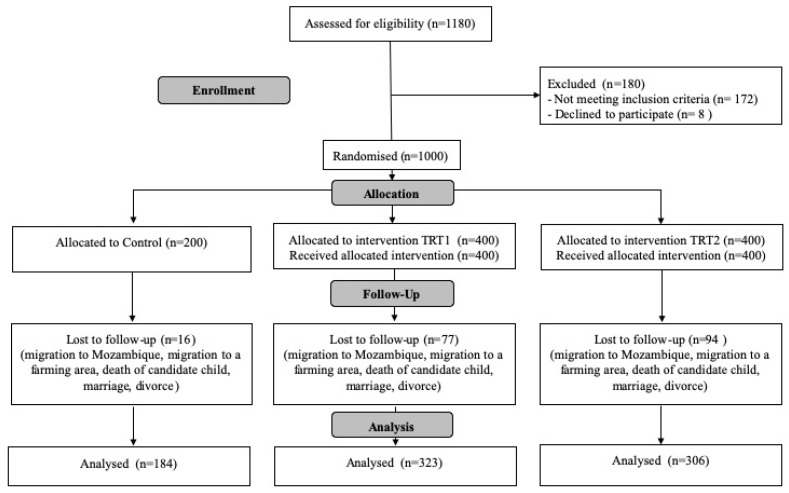
The intervention profile and analysis populations for primary outcomes (CONSORT diagram).

**Figure 3 ijerph-17-02648-f003:**
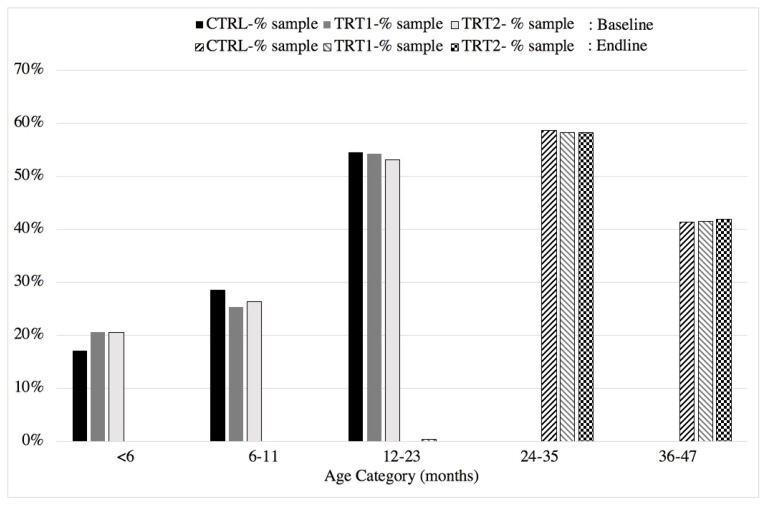
The percentage of the sample in each age category at baseline and end line of the intervention (baseline: 2017; end line: 2018).

**Figure 4 ijerph-17-02648-f004:**
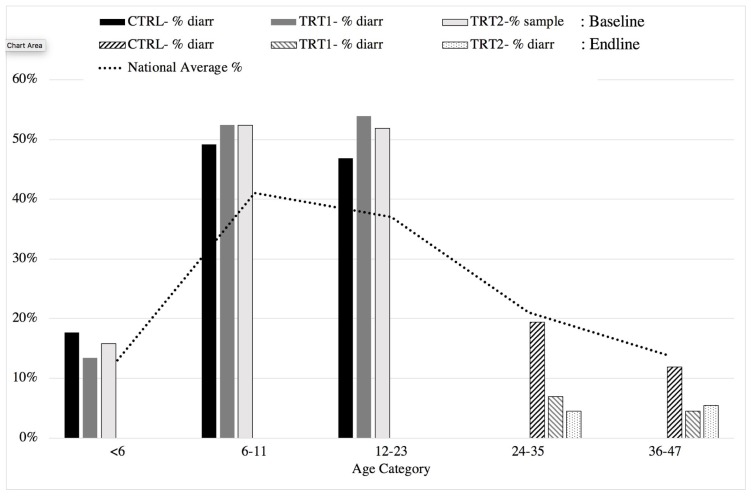
A summary of the percentage cases of diarrhoeal disease by age compared to national trends (baseline: 2017; end line: 2018).

**Figure 5 ijerph-17-02648-f005:**
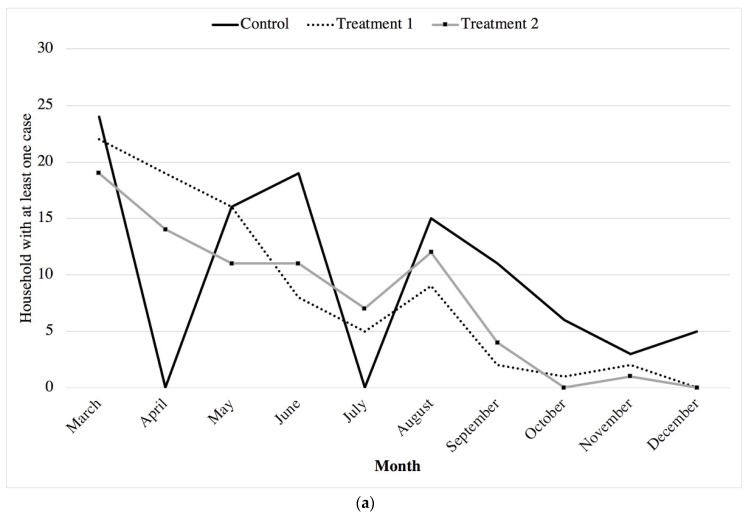
**(a)** Continuous diarrhoeal disease cases reported in terms of at least one per household (2018); (**b**) Continuous respiratory disease cases reported in terms of at least one per household (2018).

**Table 1 ijerph-17-02648-t001:** Proxy measures compared between Control and Treatment 1 at baseline (2017) and end line (2018).

Proxy Measures	Control		Treatment 1		*p*-Value
	BL % (n)	EL %(n)	BL % (n)	EL % (n)	
Presence of soap at HH	63% (125)	72% (132)	61% (243)	93% (302)	<0.001
Presence of handwashing facility (HWF)	51% (102)	35% (65)	41% (164)	98% (316)	<0.001
Presence of soap and water at HWF	24% (48)	18% (33)	21% (84)	84% (271)	<0.001
Presence of soap and water at utensil washing location	28% (56)	24% (44)	33% (132)	72% (231)	<0.001
Presence of dish rack	39% (78)	26% (48)	27% (108)	98% (316)	<0.001

Note: *n* for control at baseline: 200, at follow up: 184. *n* for treatment 1 at baseline: 400, at follow up: 323. *n* for treatment 2 at baseline: 400, at follow up: 306. Baseline (BL); End line (EL); Household (HH); Difference (diff), frequencies, chi-square.

**Table 2 ijerph-17-02648-t002:** Proxy measures compared between Control and Treatment 2 at baseline (2017) and end line (2018).

Proxy Measures	Control		Treatment 2		*p*-Value
	BL % (n)	EL %(n)	BL % (n)	EL % (n)	
Presence of soap at HH	63% (125)	72% (132)	59% (234)	93% (285)	<0.001
Presence of handwashing facility (HWF)	51% (102)	35% (65)	44% (176)	95% (291)	<0.001
Presence of soap and water at HWF	24% (48)	18% (33)	9% (36)	70% (213)	<0.001
Presence of soap and water at utensil washing location	28% (56)	24% (44)	32% (128)	68% (208)	<0.001
Presence of dish rack	39% (78)	26% (48)	30% (120)	97% (298)	0.0001

Note: *n* for control at baseline: 200, at follow up: 184. *n* for treatment 1 at baseline: 400, at follow up: 323. *n* for treatment 2 at baseline: 400, at follow up: 306. Baseline (BL); End line (EL); Household (HH); Difference (diff), frequencies, chi-square.

**Table 3 ijerph-17-02648-t003:** Proxy measures compared between Treatment 1 and Treatment 2 at baseline (2017) and end line (2018).

Proxy Measures	Treatment 1		Treatment 2		*p*-Value
	BL % (n)	EL %(n)	BL % (n)	EL % (n)	
Presence of soap at HH	61% (243)	93% (302)	59% (234)	93% (285)	0.788
Presence of handwashing facility (HWF)	41% (164)	98% (316)	44% (176)	95% (291)	0.130
Presence of soap and water at HWF	21% (84)	84% (271)	9% (36)	70% (213)	0.606
Presence of soap and water at utensil washing location	33% (132)	72% (231)	32% (128)	68% (208)	0.433
Presence of dish rack	27% (108)	98% (316)	30% (120)	97% (298)	0.280

Note: *n* for control at baseline: 200, at follow up: 184. *n* for treatment 1 at baseline: 400, at follow up: 323. *n* for treatment 2 at baseline: 400, at follow up: 306. Baseline (BL); End line (EL); Household (HH); Difference (diff), frequencies, chi-square.

**Table 4 ijerph-17-02648-t004:** Attendance at cluster meetings and participation in household visits.

			Treatment 1			Treatment 2		
		Total Held	Average Attendance	% None Attended	% All Attended	Average Attendance	% None Attended	% All Attended
Cluster meetings	Handwashing with Soap	4	2.89	7%	39%	2.76	16%	42%
	Food Safety and Hygiene	8	4.35	25%	15%	5.53	3%	19%
	Faeces Management	3				2.87	0%	89%
	Water Management	2				1.74	3%	77%
Household Visits	Handwashing with Soap	3	2.13	3%	51%	2.85	1%	88%
	Food Safety and Hygiene	7	5.90	1%	40%	6.12	1%	43%
	Faeces Management	2				1.81	1%	82%
	Water Management	2				1.74	6%	80%

**Table 5 ijerph-17-02648-t005:** Difference in differences analysis for diarrhoeal disease and respiratory infections.

	Diarrhoea		Respiratory Infection	
	Treatment 1	Treatment 2	Treatment 1	Treatment 2
	**Before**			
**Control**	0.422	0.422	0.568	0.568
**Treated**	0.452	0.448	0.560	0.617
**Diff (T-C)**	0.03	0.025	−0.008	0.050
	*(0.399)*	*(0.48)*	*(0.847)*	*(0.216)*
	**After**			
**Control**	0.158	0.158	0.29	0.290
**Treated**	0.059	0.049	0.220	0.210
**Diff (T-C)**	−0.100	−0.109	−0.07	−0.080
	*(0.010) ****	*(0.005) ****	*(0.107)*	*(0.065) **
**Diff-in-Diff**	−0.13	−0.135	−0.062	−0.129
	*(0.014) ***	*(0.011) ***	*(0.296)*	*(0.028) ***
**R^2^**	0.15	0.16	0.11	0.13

Significant at the * 10%, ** 5% and *** 1% levels. Datasets for this Table are available in Appendix A.

**Table 6 ijerph-17-02648-t006:** A linear probability model predicting outcomes at end line (*p*-values in parentheses; significant figures in bold).

	Self Reported Diarrhoea in Previous 2 Weeks				Self Reported Respiratory Infections in Previous 2 Weeks			
	(1)	(2)	(3)	(4)	(5)	(6)	(7)	(8)
**Treatment 1**		−0.096				−0.072		
		**(0.001)**				**(0.081)**		
**Treatment 2**				−0.101				−0.078
				**(0.00)**				**(0.057)**
**Diarrhoea baseline**	0.019	0.019	0.010	0.009				
	(0.498)	(0.494)	(0.701)	(0.729)				
**Respiratory baseline**					0.105	0.102	0.085	0.082
					**(0.008)**	**(0.01)**	**(0.034)**	**(0.04)**
**Visits**	−0.005		0.001		−0.016		0.017	
	(0.285)		(0.926)		**(0.032)**		**(0.055)**	
**Meetings**	−0.007		−0.008		0.007		−0.022	
	(0.153)		(0.144)		(0.343)		**(0.007)**	
**Income**	0.000	0.000	0.000	0.000	0.000	0.000	0.000	0.000
	(0.392)	(0.513)	(0.418)	(0.449)	(0.489)	(0.424)	(0.309)	(0.287)
**Gender of child**	0.009	0.004	−0.026	−0.025	−0.020	−0.025	−0.071	−0.071
	(0.746)	(0.876)	(0.312)	(0.334)	(0.601)	(0.516)	(0.069)	**(0.071)**
**Handwashing station**	−0.015	−0.017	0.045	0.046	0.031	0.020	0.019	0.017
	(0.612)	(0.561)	(0.154)	(0.15)	(0.472)	(0.636)	(0.691)	(0.717)
**Child drinks water**	−0.060	−0.055	−0.074	−0.073	0.012	0.011	−0.113	−0.116
	(0.190)	(0.230)	(0.093)	(0.096)	(0.858)	(0.862)	(0.088)	**(0.079)**
**Number of animals**	0.002	0.003	−0.010	−0.010	0.001	0.001	−0.009	−0.009
	(0.849)	(0.785)	(0.28)	(0.278)	(0.943)	(0.948)	(0.548)	(0.529)
**% vaccines accomplished**	0.005	0.016	0.086	0.080	−0.069	−0.055	−0.039	−0.050
	(0.952)	(0.839)	(0.27)	(0.302)	(0.538)	(0.628)	(0.737)	(0.672)
**Months old**	−0.003	−0.003	−0.003	−0.003	−0.001	−0.001	−0.002	−0.002
	(0.293)	(0.243)	(0.347)	(0.348)	(0.813)	(0.829)	(0.557)	(0.684)
**Have soap**	−0.020	−0.016	0.001	−0.002	−0.067	−0.060	−0.055	−0.063
	(0.465)	(0.56)	(0.971)	(0.928)	(0.102)	(0.143)	(0.175)	(0.123)
**Volume of water storage**	0.000	0.000	0.000	0.000	0.000	0.000	0.000	0.000
	(0.814)	(0.789)	(0.56)	(0.471)	(0.742)	(0.647)	(0.525)	(0.372)
**Constant**	0.219	0.214	0.169	0.179	0.313	0.292	0.447	0.466
	(0.011)	(0.014)	(0.06)	(0.044)	(0.015)	(0.023)	(0.001)	(0.001)
***n***	496	498	478	483	496	498	478	483
**R-squared**	0.044	0.039	0.063	0.060	0.034	0.027	0.055	0.042

Datasets for this Table are available in Appendix A.

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
