# Peer review of "Health Outcomes of an Integrated Behaviour-Centred Water, Sanitation, Hygiene and Food Safety Intervention–A Randomised before and after Trial"

_ijerph, 2020, doi:10.3390/ijerph17082648_

Round 1

Reviewer 1 Report

According to the Instructions for Authors, tables and figures should be inserted into the main text close to their first citation. Besides, tables and figures should have a short explanatory title and caption but, in this manuscript, they are too short.

  1. Materials and Methods

Background and Study Design

The two treatment arms were designed based on formative research: (1) food hygiene intervention only (n=400); and (2) integrated WASH and food hygiene intervention (n=400) [30] with a control group in which no intervention took place (n=200). 

According to a previous study by the authors (30) children participating were <6months, 6-11 months, and children 12-23 months.  Fig. 3 should be better explained, if age category is in months and the number of months that each category at baseline became in the other age categories at the end line. Authors only mention that the proportion of children within specific age categories was consistent across the two treatments and control areas; however, how many children <6 months age category moved to higher age categories to 12-23 category at end line of the study?

Authors analyzed the results using a difference-in-differences analysis, which is a quasi-experimental design that makes use of longitudinal data from treatment and control groups. DID is typically used to estimate the effect of a specific intervention or treatment (such as large-scale program implementation) by comparing the changes in outcomes over time between a population that is enrolled in a program (the intervention group) and a population that is not (the control group). Also, they used RANAS (Risks, Attitudes, Norms, Abilities, and Self-regulation) approach for designing and evaluating behavior change strategies that target and change the behavioral factors of a specific behavior in a specific population. However, I considered that even these analyses provide interesting results, the Relative risk should be calculated as well. Relative risk is a statistical term used to describe the chances of a certain event occurring among one group versus another. It is commonly used in epidemiology and evidence-based medicine, where relative risk helps to identify the probability of developing a disease after an exposure versus the chance of developing the disease in the absence of that exposure. It can be use in Cohort and Case-Control Studies where the objective is based on the incidence of developing a medical condition in the exposed and unexposed groups. No epidemiological data of these areas were provided at baseline and at end line study.

Author Response

Many thanks for taking the time to review our submission. Your feedback was very useful and has helped to strengthen the content. Please find below responses to your comments 

Reviewer comment

Response

According to the Instructions for Authors, tables and figures should be inserted into the main text close to their first citation. Besides, tables and figures should have a short explanatory title and caption but, in this manuscript, they are too short.

Placement and titles of tables and figures have been reviewed and updated throughout the manuscript

According to a previous study by the authors (30) children participating were <6months, 6-11 months, and children 12-23 months.  Fig. 3 should be better explained, if age category is in months and the number of months that each category at baseline became in the other age categories at the end line.

Authors only mention that the proportion of children within specific age categories was consistent across the two treatments and control areas; however, how many children <6 months age category moved to higher age categories to 12-23 category at end line of the study?

Caption of the x-axis has been updated accordingly.

In terms of the movement across age categories, as shown on Figure 3, the entire cohort of children that were in the <6, 6-12, and 12-23 age categories graduated into the 24-35 and 36-47 categories except for 1 child who remained in the 12-23 age category;  this child was from Treatment 1.

This descriptive text has been added to page 6 to provide clarification.

However, I considered that even these analyses provide interesting results, the Relative risk should be calculated as well. Relative risk is a statistical term used to describe the chances of a certain event occurring among one group versus another. It is commonly used in epidemiology and evidence-based medicine, where relative risk helps to identify the probability of developing a disease after an exposure versus the chance of developing the disease in the absence of that exposure. It can be use in Cohort and Case-Control Studies where the objective is based on the incidence of developing a medical condition in the exposed and unexposed groups.

The results presented in Table 6 are presented as percentages which are for the general reader or practitioner easier to interpret;  however, as you rightly point out, the RR values are useful, so we have attached an additional table in the Appendix that shows the logistic regression estimations as RR values

No epidemiological data of these areas were provided at baseline and at end line study.

We are unsure of the reviewers request. We feel our results outlined within this manuscript describe the population exposure levels and health effect values observed from our samples, and in the case of Figure 4 these are compared to national average health outcomes. We request further clarification of the recommendation if this response is not sufficient.

Reviewer 2 Report

Here authors have presented an excellent randomized study depicting the health outcomes of an integrated behavior centered intervention with a focus on water, sanitation, hygiene, and food safety interventions.

I have a few comments. Did authors include/analyze other variables affecting the prevalence of diarrheal and respiratory infections such as low birth weight status, malnutrition, infectious diseases (such as measles, HIV status), micronutrient deficiencies (vitamin A)? If not, this can be added as a limitation.

Author Response

Many thanks for your comments which have helped to strengthen our manuscript submission. Please find below responses to your comments. 

Reviewer comment

Response

Did authors include/analyze other variables affecting the prevalence of diarrheal and respiratory infections such as low birth weight status, malnutrition, infectious diseases (such as measles, HIV status), micronutrient deficiencies (vitamin A)? If not, this can be added as a limitation.

This is noted and has been added as a limitation on Page 6 of the manuscript.

“We also acknowledge that our study did not measure all variables which may affect the prevalence of diarrhoea or respiratory infections, such as low birth weight status, malnutrition, HIV status, or presence of other diseases such as measles, malaria, etc.”  

Reviewer 3 Report

1st Review

Title:  Health outcomes of an integrated behaviour centred water, sanitation, hygiene and food safety intervention – a randomised before and after trial

Manuscript ID: ijerph-753947

The manuscript “Health outcomes of an integrated behaviour centred water, sanitation, hygiene and food safety intervention – a randomised before and after trial” aimed to measure the relative impact of the hygiene of complementary foods on diarrhoeal disease as a primary outcome, and the impact of a food hygiene intervention (food hygiene + handwashing) relative to a food hygiene and WASH intervention (food hygiene + handwashing + faeces management + water management. The theme is interesting, but I have some minor recommendations to improve the quality of the paper.

  • Title: remove the dot from the end of the title.
  • Number the lines to facilitate when we need to mention the location of the correction.
  • Abstract: The use of a structured abstract may help people to understand the study (background, objective, methods, results, and conclusion)
  • Avoid the use of acronyms in the keywords (WASH, LMIC).
  • What is LMIC?
  • “Diarrhoeal disease continues to be a leading cause of mortality and morbidity in low and middle income countries…” – insert the mortality and morbidity data.
  • 2% - insert the “n”.
  • See the references:
    • Draeger, C.L. et al. Brazilian Foodborne Disease National Survey: Evaluating the Landscape after 11 Years of Implementation to Advance Research, Policy, and Practice in Public Health. Nutrients2019, 11, 40.
    • da Silva Farias, A. et al. Good Practices in Home Kitchens: Construction and Validation of an Instrument for Household Food-Borne Disease Assessment and Prevention.  J. Environ. Res. Public Health2019, 16, 1005.
  • “Physical recruitment was conducted by trained research assistants with the approval and support of community health workers (Health Surveillance Assistants (HSAs))…” – remove the last “)”
  • Table 1 should be placed on page 4.
  • “These surveys were conducted by trained research assistants, recorded on tablets using Kobo Collect (https://www.kobotoolbox.org), cleaned and prepared for analysis”. Use the correct way to cite the site as a reference.
  • Figure 3 was not mentioned before its appearance in the main text.
  • Page 7 – last paragraph, 8th line: insert a dot before “however”.
  • Page 3 of 21 – line 87 – cite the year after the months.
  • In all figures and tables, insert the year of data collection and the Country.
  • Line 5 of 21 – line 150 - you should insert the complete description before the acronym (RANAS).
  • Line 160 – Change the reference citation format.
  • Line 194 – insert the year after the month (November).

Author Response

Many thanks for your review and helpful comments which we have taken on board to strengthen our submission. Please find below responses to your specific comments. 

Reviewer comment

Response

·       Title: remove the dot from the end of the title

Done

·       Number the lines to facilitate when we need to mention the location of the correction

Done. Apologies has this was submitted with line numbers but some formatting seems to have changed 

Abstract: The use of a structured abstract may help people to understand the study (background, objective, methods, results, and conclusion)

As authors we followed the IJERPH instructions on abstract formatting as copied below, therefore have maintained the current abstract format.

The abstract should be a total of about 200 words maximum. The abstract should be a single paragraph and should follow the style of structured abstracts, but without headings: 1) Background: Place the question addressed in a broad context and highlight the purpose of the study; 2) Methods: Describe briefly the main methods or treatments applied. Include any relevant preregistration numbers, and species and strains of any animals used. 3) Results: Summarize the article's main findings; and 4) Conclusion: Indicate the main conclusions or interpretations.

·       Avoid the use of acronyms in the keywords (WASH, LMIC)

These have been updated 

What is LMIC?

Low and Middle Income Countries

·       “Diarrhoeal disease continues to be a leading cause of mortality and morbidity in low and middle income countries…” – insert the mortality and morbidity data

The sentence in question elaborates these figures following the section which the reviewer highlighted: “Diarrhoeal disease continues to be a leading cause of mortality and morbidity in low and middle income countries, with 62.2% of diarrhoeal disease deaths in children under five attributed to poor water, sanitation and hygiene (WASH) and the associated consumption of contaminated foods [1-3].” 

·       2% - insert the “n”

We are not sure to which part of the manuscript this is referring. Have searched for ‘2%’ and have not found an area where the “n” value is missing?

·       See the references:

·       Draeger, C.L. et al. Brazilian Foodborne Disease National Survey: Evaluating the Landscape after 11 Years of Implementation to Advance Research, Policy, and Practice in Public Health. Nutrients2019, 11, 40.

·       da Silva Farias, A. et al. Good Practices in Home Kitchens: Construction and Validation of an Instrument for Household Food-Borne Disease Assessment and Prevention.  J. Environ. Res. Public Health2019, 16, 1005.

Thanks for bringing these to our attention. We have chosen to include Draegar et al (2018) in the Introduction (reference 3). Although very useful we feel that the da Silva Farias (2019) reference was not suitable for this paper as it primarily discussed the development of a tool for measuring food hygiene practice. However we will be using this reference for a future manuscript being prepared using further data from this study.

·       “Physical recruitment was conducted by trained research assistants with the approval and support of community health workers (Health Surveillance Assistants (HSAs))…” – remove the last “)”

This has been removed 

·       Table 1 should be placed on page 4.

We are unsure why the reviewer suggests that Table 1 be moved to page 4 at the beginning of the results section. Therefore we have maintained table 1 in the same position adjacent to Tables 2 and 3 which also summarise proxy findings

·       “These surveys were conducted by trained research assistants, recorded on tablets using Kobo Collect (https://www.kobotoolbox.org), cleaned and prepared for analysis”. Use the correct way to cite the site as a reference.

The KoBo Collect software has now been cited appropriately [36] on page 4

·       Figure 3 was not mentioned before its appearance in the main text.

The text has now been moved above the Figure.

·       Page 7 – last paragraph, 8th line: insert a dot before “however”.

This has been added 

·       Page 3 of 21 – line 87 – cite the year after the months

The year (2018) has been added 

·       In all figures and tables, insert the year of data collection and the Country.

The whole study took place in Malawi and therefore the country has not been added to Figures. However we have added the years to Table and Figure titles where appropriate

·       Line 5 of 21 – line 150 - you should insert the complete description before the acronym (RANAS).

The expansion of RANAS has been added before the acronym. 

·       Line 160 – Change the reference citation format.

The citation has been updated to the correct format

·       Line 194 – insert the year after the month (November).

The year (2018) has been inserted 

Round 2

Reviewer 1 Report

The manuscript has been improved and warrants its publication.